# Postinhibitory excitation in motoneurons can be facilitated by hyperpolarization-activated inward currents: A simulation study

**Laura Schmid**[1], **Thomas Klotz**[1], **Oliver Röhrle**[1,2], **Randall K. Powers**[3], **Francesco Negro**[4◉], **Utku Ş. Yavuz**[5◉] *

1 Institute for Modelling and Simulation of Biomechanical Systems, University of Stuttgart, Stuttgart, Germany, 2 Stuttgart Center for Simulation Sciences (SC SimTech), University of Stuttgart, Stuttgart, Germany, 3 Department of Physiology and Biophysics, University of Washington, Seattle, Washington, United States of America, 4 Department of Clinical and Experimental Sciences, Università degli Studi di Brescia, Brescia, Italy, 5 Department of Biomedical Signals and Systems, Faculty of Electrical Engineering, Mathematics and Computer Sciences, University of Twente, Enschede, Netherlands

◉ These authors contributed equally to this work.
* s.u.yavuz@utwente.nl

**Data Availability Statement:** The source code and data used to produce the results and analyses

## Abstract

Postinhibitory excitation is a transient overshoot of a neuron's baseline firing rate following an inhibitory stimulus and can be observed *in vivo* in human motoneurons. However, the biophysical origin of this phenomenon is still unknown and both reflex pathways and intrinsic motoneuron properties have been proposed. We hypothesized that postinhibitory excitation in motoneurons can be facilitated by hyperpolarization-activated inward currents (h-currents). Using an electrical circuit model, we investigated how h-currents can modulate the postinhibitory response of motoneurons. Further, we analyzed the spike trains of human motor units from the tibialis anterior muscle during reciprocal inhibition. The simulations revealed that the activation of h-currents by an inhibitory postsynaptic potential can cause a short-term increase in a motoneuron's firing probability. This result suggests that the neuron can be excited by an inhibitory stimulus. In detail, the modulation of the firing probability depends on the time delay between the inhibitory stimulus and the previous action potential. Further, the postinhibitory excitation's strength correlates with the inhibitory stimulus's amplitude and is negatively correlated with the baseline firing rate as well as the level of input noise. Hallmarks of h-current activity, as identified from the modeling study, were found in 50% of the human motor units that showed postinhibitory excitation. This study suggests that h-currents can facilitate postinhibitory excitation and act as a modulatory system to increase motoneuron excitability after a strong inhibition.

## Author summary

Human movement is determined by the activity of specialized nerve cells, the motoneurons. Each motoneuron activates a specific set of muscle fibers. The functional unit consisting of a neuron and muscle fibers is called a motor unit. The activity of motoneurons

presented in this manuscript are available from https://doi.org/10.18419/darus-3686.

**Funding:** This work was supported by the Deutsche Forschungsgemeinschaft (DFG, German Research Foundation, SPP 2311 (465243391) to LS) and the European Research Council (ERC-AdG "qMOTION" (Grant agreement ID: 101055186) to TK; Consolidator Grant INcEPTION (contract no. 101045605) to FN). The funders had no role in study design, data collection and analysis, decision to publish, or preparation of the manuscript.

**Competing interests:** The authors have declared that no competing interests exist.

can be observed noninvasively in living humans by recording the electrical activity of the motor units using the electromyogram. We studied the behavior of human motor units in an inhibitory reflex pathway and found an unexpected response pattern: a rebound-like excitation following the inhibition. This has occasionally been reported for human motor units, but its origin has never been systematically studied. In non-human cells of the neural system, earlier studies reported that a specific membrane protein, the so-called h-channel, can cause postinhibitory excitation. Our study uses a computational motoneuron model to investigate whether h-channels can cause postinhibitory excitation, as observed in the experimental recordings. Using the model, we developed a method to detect features of h-channel activity in human recordings. Because we found these features in half of the recorded motor units, we conclude that h-channels can facilitate postinhibitory excitation in human motoneurons.

## Introduction

Sherrington noted in his 1909 studies that "reflex inhibition of the vastocrureus and other extensor muscles in decerebrate rigidity is followed regularly, under certain circumstances, on withdrawal of the inhibitory stimulus, by a rebound contraction" [1]. Interestingly, the "rebound contraction" persisted after de-afferentation [1]. Later, the same phenomenon of motor unit postinhibitory excitation was reported in several studies [2–6]. The proposed mechanisms include reflex pathways activated by muscle spindles and intrinsic neuron characteristics originating from the behavior of specific ion channels. However, none of these studies has systematically investigated the origin of the phenomenon.

Similar to Sherrington's findings, we observed a consistent rebound-like excitation following electrically evoked reciprocal inhibition in human tibialis anterior motor units (Fig 1c) [7]. Reciprocal inhibition is a component of the stretch reflex pathway, which is disynaptically elicited by muscle spindles of an antagonistic muscle (Fig 1a) [8, 9]. Here, reciprocal inhibition was superimposed on sustained isometric contractions of the tibialis anterior muscle by electrical stimulation of the tibial nerve.

Increased excitability elicited by a purely inhibitory stimulus is a well-known phenomenon in different cells of the mammalian neural system, and various ion channels were found to cause or facilitate this behavior *in vitro* [10–14]. Therefore, it is surprising that intrinsic motoneuron mechanisms are rarely considered to explain the postinhibitory increase in activity of motor units *in vivo*. Instead, excitatory synaptic inputs based on reflex pathways were lately considered [2, 3, 5, 6].

Ito and Oshima [11] first described membrane potential overshoots caused by hyperpolarization-activated inward currents in cat spinal motoneurons. Takahashi [15] found that this current, which is commonly named h-current or $I_h$, is mediated by sodium and potassium ions. The corresponding channel family was identified as hyperpolarization-activated cyclic nucleotide-gated non-selective cation channels (HCN channels), which are expressed throughout the soma and dendrites of spinal motoneurons [16]. Here, we refer to this group of channels as h-channels and the corresponding currents as h-currents.

Based on the large body of evidence for intrinsic postinhibitory excitation mechanisms from *in-vitro* studies, we hypothesized that intrinsic motoneuron mechanisms, specifically h-currents, also contribute to the postinhibitory excitation in motor units observed *in vivo* (Fig 1b). In this study, we analyzed for the first time the incidence and amplitude of postinhibitory excitation in human motor units following electrically elicited reciprocal inhibition of the

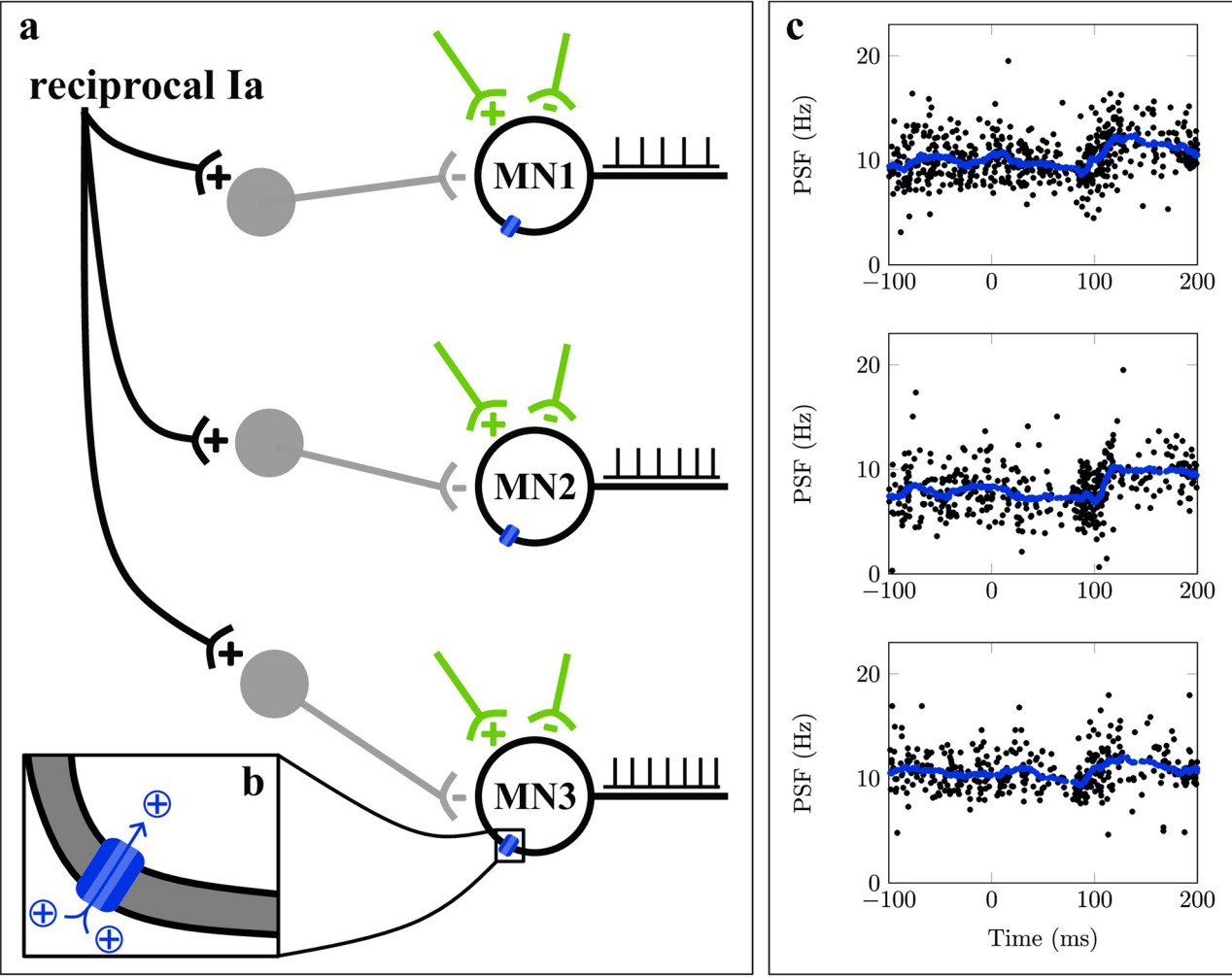

**Fig 1. Postinhibitory excitation in human motor units.** (a): Motoneurons (MNs) receive multiple, typically unknown synaptic inputs (green). Particularly, reciprocal inhibition is mediated by interneurons (gray). In humans, the recording of motor unit spike trains (black) allows for studying the function of motoneurons. (b): The integration of synaptic inputs in motoneurons is determined by ion channels (blue). (c): Peristimulus frequencygram (PSF) of three exemplary tibialis anterior motor units in response to reciprocal inhibition, elicited by electrical stimulation of the tibial nerve during sustained isometric contractions. Moving average is shown in blue, data from [7]. It is unclear if the observed postinhibitory excitation is caused by neural pathways or intrinsic motoneuron properties.

tibialis anterior muscle. Then we used a computational motoneuron model to reproduce the experimental results and understand the role of h-currents in postinhibitory excitation. To this end, we employed a compartmental electric circuit model, which is based on previous works by Cisi and Kohn [17], Negro and Farina [18] and Powers et al. [19]. We analyzed and compared the discharge behavior of both *in-silico* and *in-vivo* motoneurons using the peristimulus frequencygram (PSF) [20].

## Results

First, we will describe the postinhibitory excitation pattern observed in human motor units. Then, we will analyze the conditions under which we observed postinhibitory excitation in the simulated motoneuron. In the simulated motoneuron, we will determine the hallmarks of h-

current-mediated postinhibitory excitation. The insights gained will be used to analyze the experimental data for possible evidence of hyperpolarization-activated inward current activity.

## Postinhibitory excitation in human motor units

We examined the incidence and amplitude of postinhibitory excitation in 159 tibialis anterior motor units identified during the reciprocal inhibition experiment, i.e., electrical stimulation of the tibial nerve during sustained isometric contractions. Therefore, we analyzed the peristimulus frequencygram (PSF), which shows the instantaneous discharge rates of motor units relative to the stimulus time [20]. The amplitude of inhibition and excitation responses was determined from the cumulative summation of the PSF (PSF-cusum) as described in Section Data analysis. PSF and PSF-cusum of an exemplary motor unit are shown in Fig 2. In the

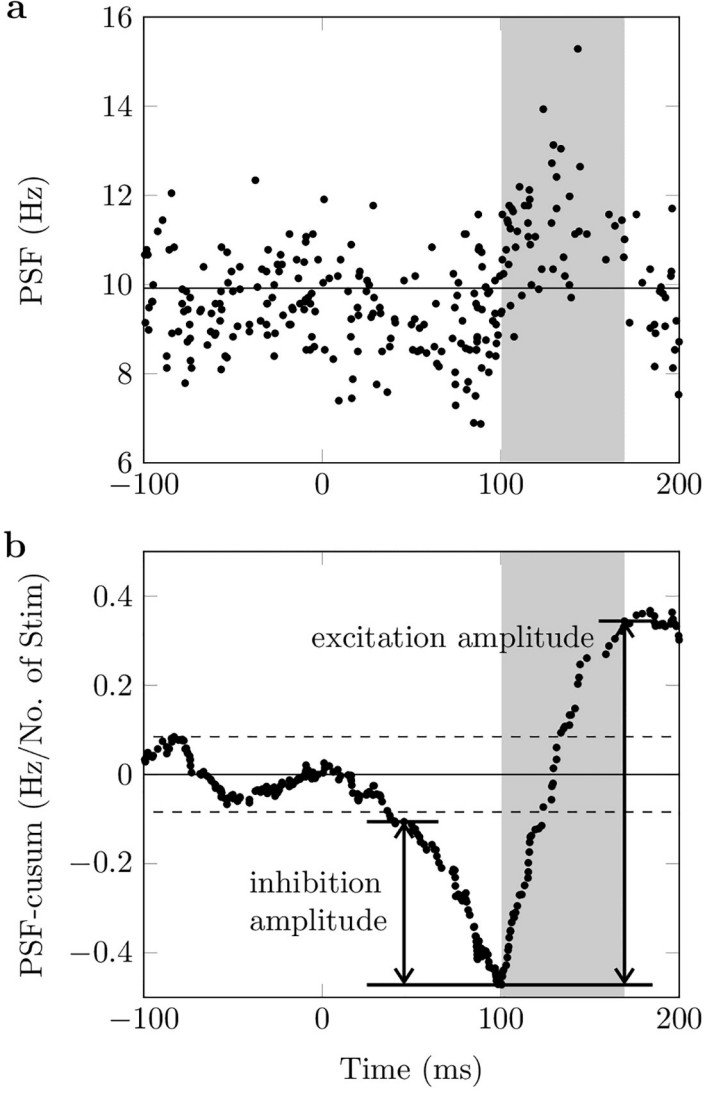

**Fig 2. Peristimulus analysis for an exemplary selected tibialis anterior motor unit.** (a): Peristimulus frequencygram (PSF). (b): Cumulative summation of PSF (PSF-cusum). The electrical stimulus to the tibial nerve was applied at time zero. Solid horizontal lines show prestimulus mean values and dashed lines mark the significance threshold for reflex responses. Arrows show the distance between manually determined turning points in PSF-cusum, i.e., inhibition and excitation amplitude. The period of postinhibitory excitation is highlighted with gray color. Data from [7].

prestimulus period, the PSF-cusum is characterized by oscillations around zero (Fig 2b). The inhibitory response is indicated by a persistent drop in PSF-cusum following the stimulus, which was applied at time zero. Notably, the time delay between the electric stimulus and the onset of the inhibition is consistent with the axonal action potential conduction velocity. A subsequent sustained increase in PSF-cusum indicates postinhibitory excitation. In total, 159 motoneurons of 9 subjects were examined. All examined motor units had a significant inhibition response, and the mean inhibition amplitude was 0.67 ± 0.47 Hz/No. 89 motor units showed significant postinhibitory excitation and the mean amplitude of the postinhibitory excitation was 1.80 ± 2.08 Hz. The mean baseline discharge rate of the analyzed motor units was 10.12 ± 1.66 Hz.

## Postinhibitory excitation in simulated motoneurons

The results for one exemplary reciprocal inhibition simulation are summarized in Fig 3. Thereby, we directly compared the behavior of the motoneuron model with h-current present (Fig 3a) and the h-current knock-out model (Fig 3b). Before the application of the inhibitory postsynaptic current (IPSC), both simulated neurons show a stationary baseline activity, i.e., stable mean discharge rate and interspike interval variability. Both motoneurons show an inhibitory response, consisting of a silent period followed by a period of reduced discharge rates, which is apparent by a decrease in PSF-cusum (Fig 3c and 3d). Although the same IPSC was applied to both computational neurons, the inhibition amplitude is larger in the neuron without h-current (1.11 Hz/No. of Stim vs. 0.34 Hz/No. of Stim). After the inhibition, only the neuron with h-current shows a significant excitation response (amplitude 0.33 Hz/No. of Stim).

To investigate the role of different factors in postinhibitory excitation in more detail, we simulated reciprocal inhibition of the motoneuron with the h-current under various conditions, i.e., varying the amplitude of the IPSC, the mean drive and the input noise level. Modifying the mean drive and noise level yielded baseline frequencies between 9.99 Hz and 18.52 Hz and a coefficient of variation of the interspike interval ranging from 0% to 12%, respectively (Table 1). Notably, it is observed that increasing the noise also yields a higher baseline discharge rate. This can be explained by the fact that higher random oscillations of the membrane potential are more likely to hit the depolarization threshold and, therefore, trigger action potentials at higher firing rates.

For all conditions, the simulated motoneuron showed an inhibition response with inhibition amplitudes ranging from 0.26 Hz/No. of Stim to 1.67 Hz/No. of Stim. However, an excitation response was not always observable. The simulated motoneuron showed postinhibitory excitation amplitudes between 0 Hz/No. of Stim, i.e., no significant excitation response, and 0.64 Hz/No. of Stim. Fig 4 summarizes the results from all simulations. It can be observed that the excitation amplitude increases with the size of the IPSC. When fixing both the mean drive and the noise level, the relationship between IPSC size and the excitation amplitude can be approximated with a linear regression model, especially for no/low noise and low/medium drive ($98\% < R^2 \leq 99.9\%$, Table 1). For a fixed IPSC size, the excitation amplitude was higher for a lower drive, i.e., smaller baseline firing rates (Table 1). Generally, the amplitude of the postinhibitory excitation decreased with increasing noise and disappeared under high noise and medium to high drive conditions.

In summary, postinhibitory excitation could only be observed in simulated neurons with hyperpolarization-activated inward currents. Systematically varying the model parameters revealed that the postinhibitory excitation amplitude is correlated with the strength of the IPSC. Yet, the excitation amplitude is modulated by the mean drive and the noise level. That

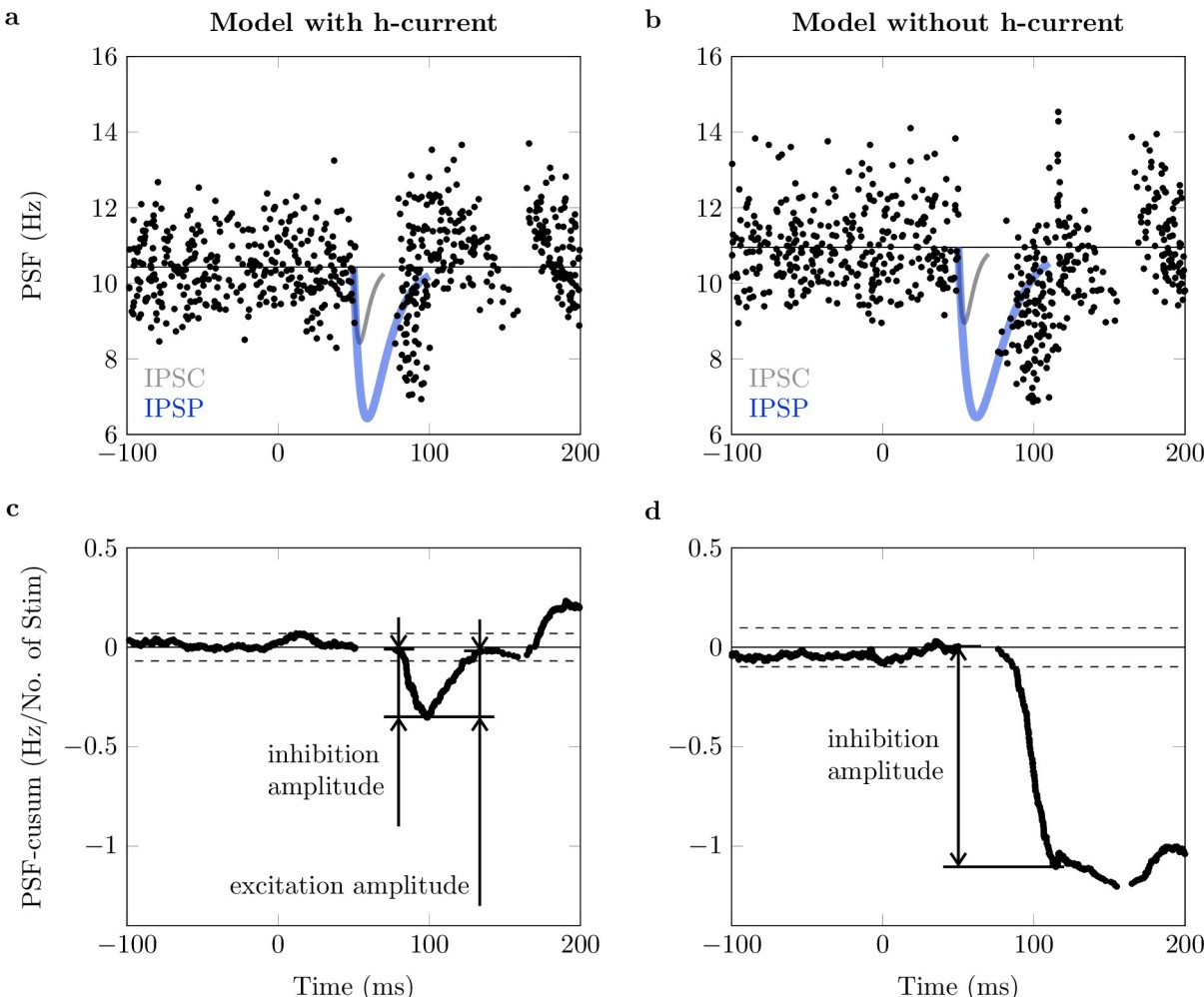

**Fig 3. Peristimulus analysis for simulated motoneurons.** (a): Peristimulus frequencygram (PSF) for a simulated neuron with h-current. (b): PSF for a simulated neuron without h-current. In (a) and (b), the injected inhibitory postsynaptic current (IPSC, amplitude -10nA) and the schematic trajectory of the induced inhibitory postsynaptic potential (IPSP) are depicted in gray and blue color, respectively. The actual time course of the membrane potential depends on the membrane potential value and the size of other inputs at IPSC application time. (c): PSF cumulative summation (PSF-cusum) for a simulated neuron with h-current. (d): PSF-cusum for a simulated neuron without h-current. Solid horizontal lines show prestimulus mean values and dashed lines mark the significance threshold for reflex responses. Arrows show the distance between two manually determined turning points in PSF-cusum, i.e., inhibition and excitation amplitude.

is, increasing the baseline firing rate or noise decreases the excitation amplitude, even to a degree where excitation is no longer observed, at low noise and high drive or high noise and medium to high drives.

## Postinhibitory excitation is stimulus-time dependent

We made use of the *in-silico* model to mechanistically link the observed postinhibitory excitation responses and the biophysical behavior of motoneurons. This was possible because the simulation allows us to observe all internal system parameters, e.g., the membrane potential trajectory, and relate them to each other. The effect of hyperpolarization-activated inward currents was isolated by comparing the trajectories of the membrane potential of computational motoneurons with and without h-channel. Here, we considered an exemplary simulation with

**Table 1. Results of the linear regression analysis.** Mean frequency and coefficient of variation of the interspike interval (CoV ISI) of the baseline activity as well as slope and coefficient of determination ($R^2$) values of the linear regression for nine different combinations of mean drive and noise input are given. Linear regression was performed using the least-squares method, provided that more than one data point was available (otherwise marked with n.a.).

| Noise | Drive | Baseline frequency (Hz) | CoV ISI (%) | Slope (Hz/No. of Stim/nA) | $R^2$ |
|---|---|---|---|---|---|
| No | Low | 9.994 | 0.048 | 0.042 | 0.998 |
| | Medium | 13.995 | 0.069 | 0.023 | 0.997 |
| | High | 17.994 | 0.079 | 0.013 | 0.993 |
| Low | Low | 10.431 | 8.718 | 0.03 | 0.981 |
| | Medium | 14.153 | 6.395 | 0.02 | 0.999 |
| | High | 18.026 | 5.98 | n.a. | n.a. |
| High | Low | 11.255 | 12.123 | 0.011 | 0.983 |
| | Medium | 14.602 | 10.166 | n.a. | n.a. |
| | High | 18.53 | 10.108 | n.a. | n.a. |

low drive and an IPSC amplitude of -10 nA and a duration of 20 ms. To focus on the effect of the h-current, we also omitted noise in these simulations.

The simulations revealed that when applying an inhibitory current pulse the instantaneous discharge rate of the simulated motoneurons always depends on the time delay between the stimulus-induced inhibitory postsynaptic potential (IPSP) and the previous motoneuron discharge $t_{stim}$ (Fig 5). Particularly, the model with h-current showed two opposing responses (Fig 5a and 5b). For large $t_{stim}$ values (Fig 5a, green trace), the next action

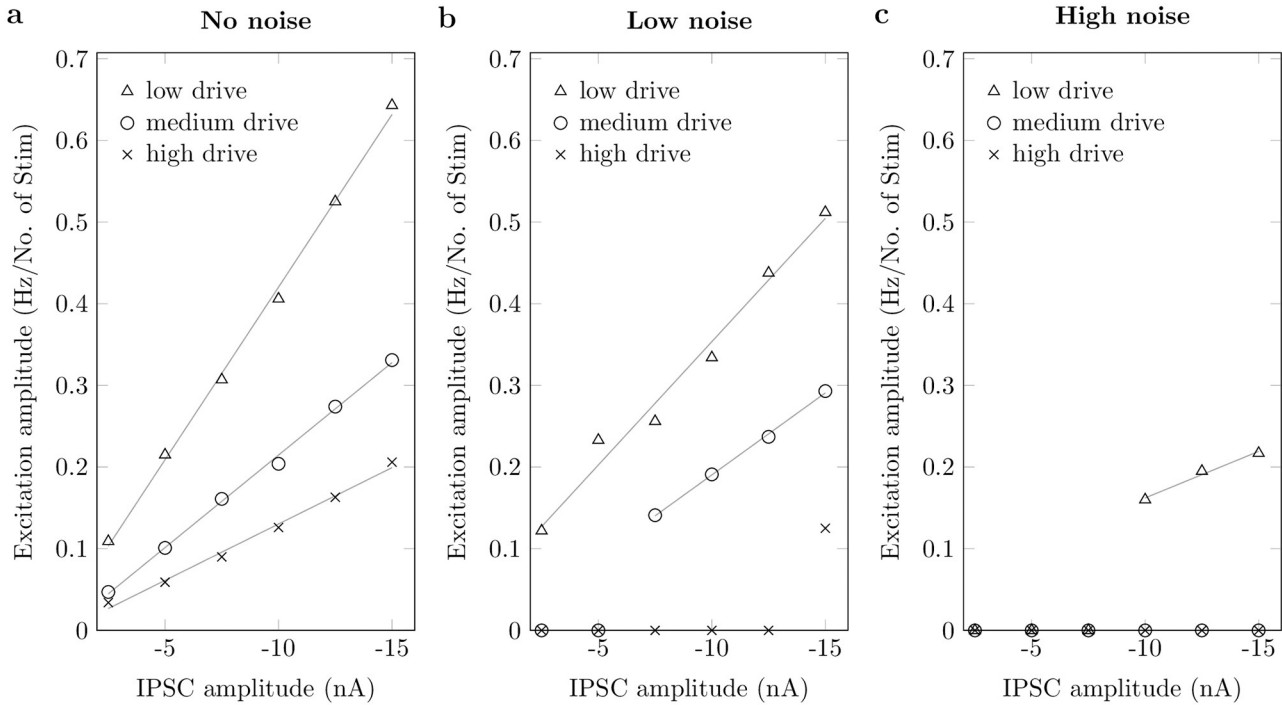

**Fig 4. Postinhibitory excitation in different simulation settings.** Excitation amplitudes in relation to inhibitory postsynaptic current (IPSC) amplitude and for three different baseline discharge rates (low (△), medium (o), and high (x) drive) and with three different amounts of noise (standard deviation 0% (a), 12.5% (b) and 25% (c) of mean drive). Gray lines show linear regressions.

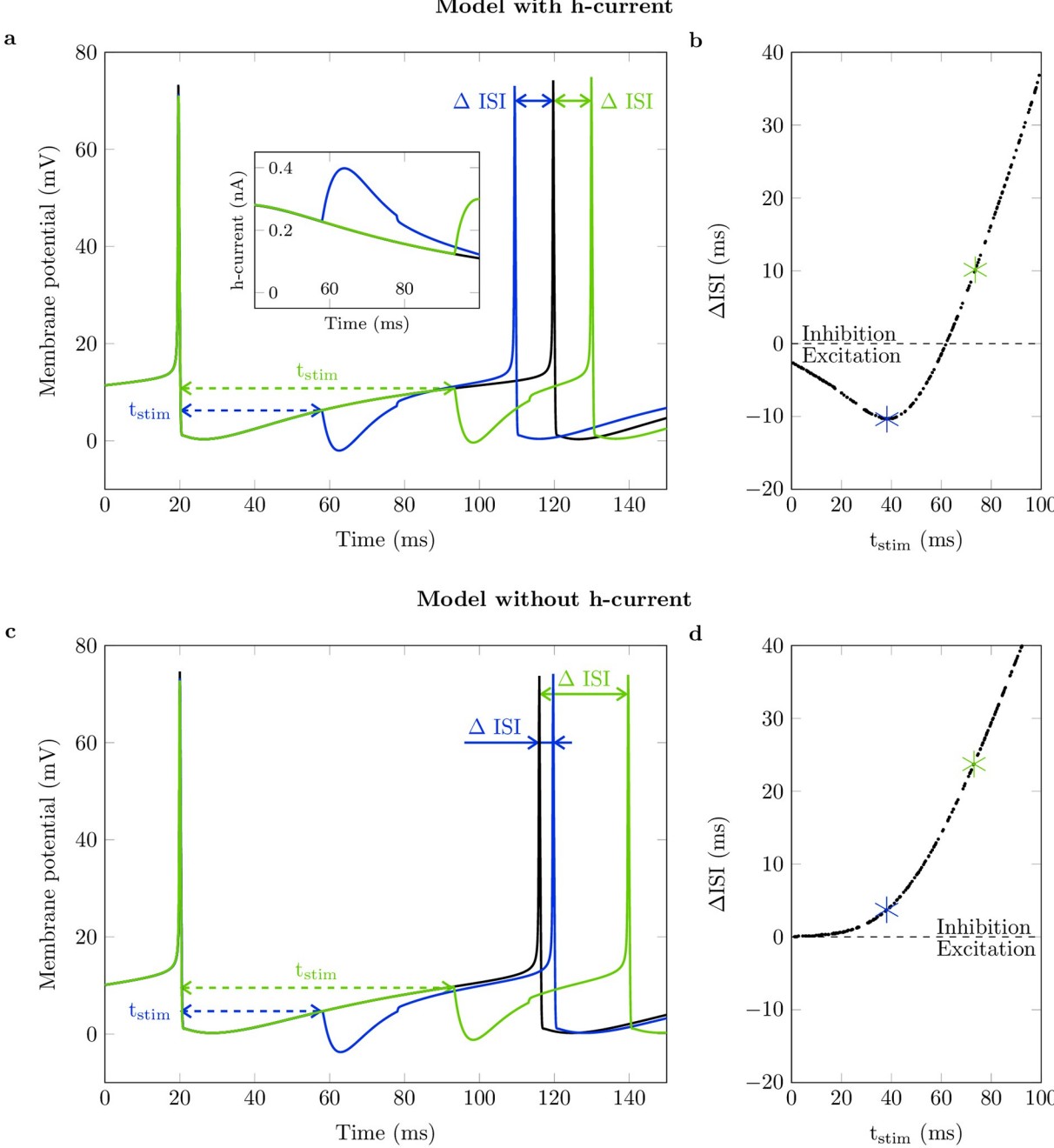

**Fig 5. Analysis of history-dependent interspike interval duration in simulated motoneurons.** Top row: model with h-current, bottom row: model without h-current. Baseline frequency 10 Hz, no noise, inhibitory postsynaptic current (IPSC) amplitude -10 nA. (a, c): Membrane potential trajectory without stimulus (black, undisturbed interval) and with stimulus applied at two exemplary time points (blue, green). Dashed arrows mark IPSC application time with respect to the last discharge ($t_{stim}$) and solid arrows mark change of interspike interval with respect to the undisturbed interval ($\Delta$ ISI). Insert in (a) shows h-current for the shown interspike intervals between 40 ms and 100 ms. Here, a positive sign indicates current flux into the cell. (b, d): Change of interspike interval duration ($\Delta$ ISI) over time of IPSC application with respect to the last discharge ($t_{stim}$). Intervals shown in (a) and (c), respectively, are marked with asterisks. Dashed lines separate prolonged interspike intervals (inhibition) from shortened interspike intervals (excitation).

potential was delayed compared to an unperturbed reference simulation (Fig 5a, black trace), i.e., the neuron was inhibited. In contrast, for a shorter $t_{stim}$ (Fig 5a, blue trace), the inter-spike interval was shortened compared to the undisturbed reference simulation, i.e., the neuron was excited. Thereby, the amount of postinhibitory excitation depends on the exact timing of the IPSP with respect to the previous spike. The insert in Fig 5a shows that the IPSP induced a positive inward ionic current flux that continued even after the end of the IPSP. Thus, the additional depolarization (blue line above the unperturbed black line) is caused by the h-current. Excitation in response to inhibitory stimuli was observed only in simulations with h-current. In the model without h-current, the IPSP always inhibited the neuron. Thereby, later IPSPs that prolonged the interspike interval (large $t_{stim}$) caused a stronger inhibition than earlier IPSPs (Fig 5c and 5d).

This characteristic time-dependent behavior of the computational motoneuron with h-current can also be visualized in the previously shown PSF and PSF-cusum plots. Therefore, the spike trains shown in the PSF plots were clustered into two groups. In detail, we separated spike trains where the instantaneous discharge frequency of the first poststimulus spike was significantly (by more than one standard deviation) higher or lower than the baseline discharge rate. Fig 6 shows the same data as previously reported in Figs 2 and 3. However, spike trains where the first postinhibitory spike shows significant excitation are visualized in blue color (excitation cluster). Spike trains where the first postinhibitory spike shows significant inhibition are shown in green color (inhibition cluster). The results of an exemplary simulated motoneuron with h-current are shown in Fig 6b and 6e and clearly show distinct excitation and inhibition clusters. This is also evident from the PSF-cusums of the clustered spike trains (Fig 6h). The initial decrease in overall PSF-cusum (black) is caused by the inhibition cluster (green). As expected, the timing of the prestimulus spikes indicates that the IPSP was delivered late in the motoneuron's afterhyperpolarization period (large $t_{stim}$, cf. Fig 5). The initial inhibition is followed by the excitation response (blue) that ultimately predominates the overall PSF-cusum. The longer latency of the spikes associated with the excitation cluster, as well as the pattern of the prestimulus spikes, are in agreement with our finding that excitation is observed when the IPSP is delivered early in the afterhyperpolarization period of the motoneuron (small $t_{stim}$). In conclusion, it is possible to find evidence for hyperpolarization-activated inward current activity by means of PSF analysis.

Hence, we investigated experimentally recorded motor units that showed significant postinhibitory excitation with the described cluster-based PSF analysis. In total, 45 of 89 motor units with significant postinhibitory excitation showed frequencies significantly higher than the baseline frequency in the first poststimulus interval (to at least 10% of delivered stimuli). The results for an exemplary chosen experimentally recorded motor unit are shown in Fig 6a, 6d and 6g. Indeed, one can observe the same characteristic behavior as for the simulated motoneuron with h-current, i.e., distinct inhibition and excitation clusters, whereby the inhibition precedes the excitation.

For completeness, the analysis was also performed for the simulated neuron without h-current. As expected, for the neuron without h-current the number of samples in the excitation cluster is very low and does not cause a relevant excitation response (Fig 6c, 6f and 6i).

From the analysis of the membrane potential and PSF of the simulated motoneurons, it can be concluded that h-currents cause the motoneuron to be either inhibited or excited following an IPSP, depending on the timing of the IPSP with respect to the last discharge. This characteristic response pattern was also found in the first poststimulus interval of half of the recorded motor units.

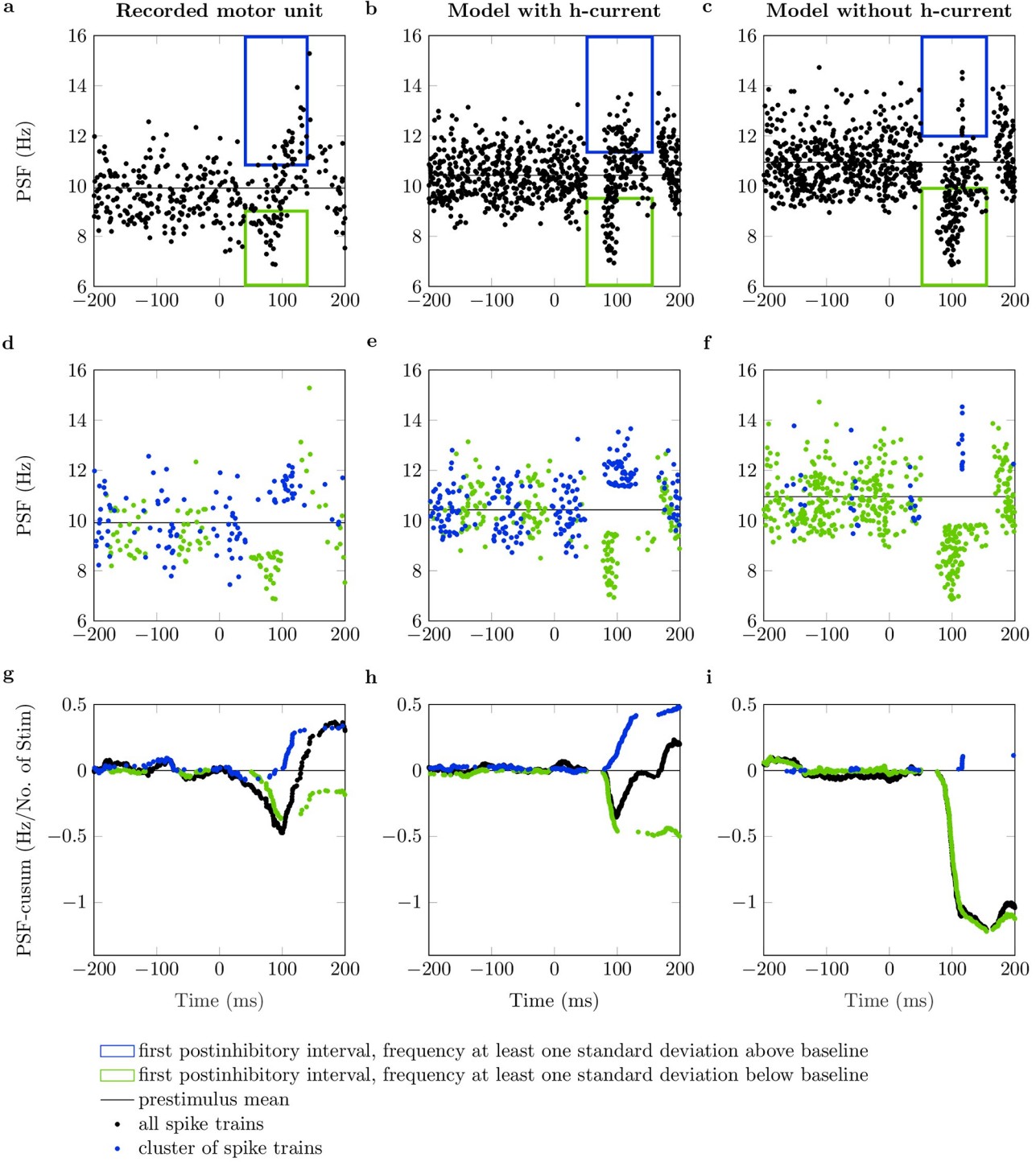

**Fig 6. Cluster-based analysis of peristimulus frequencygram (PSF).** PSF and PSF cumulative summation (PSF-cusum) for one experimentally recorded motor unit and a simulated neuron with and without h-current (data from Figs 2 and 3). The blue boxes in panels (a, b, c) cluster the first postinhibitory spikes that fire at least one standard deviation above the mean baseline frequency (black line). Accordingly, the green boxes cluster all first postinhibitory spikes that fire with at least one standard deviation below the mean baseline frequency. (d, e, f): Clusters of spike trains from which the first poststimulus spikes appear in the blue or green box, respectively. (g, h, i): PSF-cusum of all spike trains (black) and clusters of spike trains (blue, green).

## Discussion

We identified an excitatory response that frequently follows a strong reciprocal inhibition in motor units of the human tibialis anterior muscle. For the first time in motor units recorded *in vivo*, we investigated if an intrinsic motoneuron property can cause this phenomenon. Particularly, we hypothesized that a hyperpolarization-activated inward current (h-current) may be one of the factors that contribute to this excitatory response pattern. Using a computational motoneuron model, we showed that the h-current could lead to an excitation response after an inhibitory stimulus when; 1) the IPSP is applied at an appropriate time window in the neuron's integration phase, 2) the IPSP has a sufficiently large amplitude and 3) other inputs to the neuron are small. Further, it was shown that evidence for hyperpolarization-activated inward current mediated postinhibitory excitation can be tested *in vivo* using cluster-based PSF analysis. The integrated evaluation of both *in-vivo* and *in-silico* data presented within this study show that h-currents could be a mechanism that facilitates postinhibitory excitation in motoneurons. This challenges an established paradigm that postinhibitory excitation in motoneurons is dominantly caused by reflex pathways and underlines that intrinsic motoneuron properties have to be considered when interpreting *in-vivo* reflex experiments.

### Insights on mechanisms of postinhibitory excitation

A detailed analysis of the *in-silico* experiments showed that h-currents cause a non-linear, history-dependent input-output relation for short inhibitory stimuli. That is, excitation or inhibition depending on the delay between the applied IPSP and the previous motoneuron discharge. H-currents are activated at hyperpolarized membrane potentials [21]. Counterintuitively, an IPSP can lead to an increased net current influx into the cell. If the IPSP is applied when the motoneuron is close to its depolarization threshold, the instantaneous effect of the IPSP is dominant. Hence, one will observe inhibition. However, if the motoneuron is inhibited earlier in the integration phase, the IPSP-induced additional h-current influx ultimately dominates the IPSP and causes excitation.

Through a cluster-based PSF analysis, the opposing motoneuron responses can be visualized for both *in-silico* and *in-vivo* data. Thereby, spike trains are grouped according to the relative instantaneous discharge rate of the first poststimulus discharge. It was shown that a considerable fraction of the experimentally recorded motor units showed characteristic clusters of both excitation and inhibition. The same patterns were observed for the simulated motoneurons with h-currents. Hence, these findings suggest h-current activity in human motor units.

The model predicted that the excitation response negatively correlates with a motoneuron's firing rate and can even become undetectable at high noise levels. In this work, the 'high noise' condition was characterized by a coefficient of variation of the interspike interval ranging from 10% to 12%. This is at the lower end of what was reported for humans [22, 23]. This potentially explains our finding that the excitation response was not observable in all investigated motor units. Long inhibitory stimuli in combination with high discharge rates might have also prevented Türker and Powers [4] from observing excitation in the first poststimulus interval when they used a comparable cluster-based PSF analysis to investigate the effect of large inhibitory postsynaptic potentials in rat hypoglossal motoneurons.

Notably, other factors may modulate h-current mediated postinhibitory excitation. For example, persistent inward currents (PICs) can modulate h-current activity [24]. Adding a fast and a slow PIC to the motoneuron model, we could still observe history-dependent excitation (S3b Fig). Even though PICs were shown to play a minor role during Ia reciprocal inhibition [25], their role in postinhibitory excitation should be investigated in future studies.

Postinhibitory excitation of motoneurons is a well-known phenomenon that was repeatedly observed both in living humans and for *in-vitro* preparations [4–6]. Yet, no consensus regarding the biophysical origin of this phenomenon has been firmly established. Previously proposed mechanisms include intrinsic motoneuron properties like summation effects of the ionic channel conductances that are active during the afterhyperpolarization [4] or neuronal pathways, e.g., Ia and II stretch reflexes [5, 6]. In the presented simulations, we did not consider other neuronal pathways that potentially contribute to the postinhibitory excitation. Importantly, the results shown in this work only consider the first poststimulus discharge. On this time scale, the involvement of a neural pathway is unlikely. Nevertheless, we cannot exclude that the electrical stimulation of the nerve excites additional pathways, which cause an excitatory postsynaptic potential with a short delay and reinforce the postinhibitory excitation. Regarding other internal mechanisms, we could observe a mild effect of afterhyperpolarization summation in the simulations that leads to increased discharge rates of the second poststimulus spike. Further, additional simulations with a longer time constant for the h-channel showed that, under certain circumstances, the h-current could last long enough to slightly increase the frequency of the second poststimulus discharge of the neuron. Due to the uncertainty of the parameters, we refrain from a quantitative comparison with the experimental data. Nevertheless, the notably higher excitation amplitudes in the experimentally recorded motor units indicate that h-currents are not the only mechanism playing a role in the postinhibitory excitation phenomenon.

We conclude that h-currents can facilitate postinhibitory excitation observed at the first poststimulus discharge [5, 7]. However, other factors need to be considered to fully uncover the complex behavior of motoneurons in response to an IPSP. For example, postinhibitory excitation at the second poststimulus discharge [4, 6] cannot be fully explained by h-currents.

## Limitations

In this study, we used a computer model to assist the interpretation of empirical observations from reflex experiments in living humans. However, it is currently not feasible to measure all model parameters that would be needed to directly replicate the corresponding *in-vivo* experiment. Thus, the presented simulations are a simplification of the underlying physiology.

The utilized model reduces the structure of the dendritic tree into a single model compartment and does not explicitly describe all channels found in human motoneurons. Yet, previous studies showed that a two-compartment model with a limited number of conductances can replicate realistic motoneuron firing patterns [17, 18]. Adding h-channels yields the simplest model to address the posed research question sufficiently. The h-channel modulates the rheobase of the simulated motoneuron. To compare the model with h-currents and the knock-out model, the input current was adjusted, i.e., both models operate at comparable points of their current-frequency relation. This guarantees that differences are exclusively attributed to h-currents. Specifically, the membrane potential and the ion channel gating variables deviate from each other by less than 10% on average for identically long interspike intervals (S1 Fig). The parameterization of the h-channel can reproduce the magnitude of membrane potential overshoots in response to hyperpolarizing currents steps as observed *in vitro* [12, 26]. To compensate for the temperature difference between the *in-vitro* experiment and *in-vivo* conditions we followed a time-dynamics correction proposed by Powers et al. [19]. Although it is impossible to validate the utilized implementation directly, additional simulations showed that varying the h-channel parameters or motoneuron size does not affect the general behavior of the model (S2 and S3a Figs).

Reciprocal inhibition indirectly stimulates the motoneurons through the afferent nerve. The magnitude of the inhibitory input to the motoneuron caused by interneurons is unknown. To compare the simulations and experimental data, the amplitude of the injected current was chosen such that the inhibition amplitudes and latencies determined from the PSF-cusum are in the same range as for the experimental study. In this way, we assume that the injected current produces an IPSP comparable to the *in-vivo* conditions. Nevertheless, a quantitative comparison of the postinhibitory excitation amplitudes of *in-silico* and *in-vivo* motoneurons is beyond the scope of this study.

Further, instead of explicitly modeling synapses the input signals represent effective synaptic currents, i.e., currents that eventually reach the soma. This is a reasonable simplification since only these currents affect the generation of action potentials [27]. Still, injecting the inhibitory stimuli into the soma bypasses h-channels located on the dendrite to a certain extent and, hence, underestimates the effect of h-channels. Synapses for reciprocal inhibition are located close to the soma [28–30], and thus, the thereby introduced error is assumed to be small.

Postinhibitory excitation is a phenomenon that was repeatedly shown *in vitro* in different species and cell types and attributed to different mechanisms, e.g., h-channels, (calcium-activated) potassium conductances, low threshold sodium conductances, delayed recovery of the sodium inactivation gate (anode break excitation), T-type calcium channels, or NMDA receptors [12–14, 26, 31–33]. One particular strength of computer models is the possibility of studying the influence of an isolated phenomenon. The family of h-channels is a likely candidate as its gating variable shows steep slopes for hyperpolarized membrane potentials, and its time constant allows it to react at the timescale of one interspike interval [19, 21, 34]. Note that anode break excitation can be excluded in our model since the sodium inactivation gate is almost fully open at resting potential (S1c Fig). We cannot rule out the possibility that one or several of these channel mechanisms may also contribute to the postinhibitory excitation in motor units. Nevertheless, it highlights that internal motoneuron properties need to be considered to investigate postinhibitory excitation in human motor units.

### Functional significance and future directions

We showed that h-current mediated postinhibitory excitation is most pronounced when a motoneuron operates close to its recruitment threshold and for strong inhibitory stimuli. This is interesting regarding the functional significance of h-currents. We speculate that h-currents temporally increase the excitability of a motoneuron and potentially protect a motor unit from derecruitment. In previous studies, h-currents were shown to increase the excitability of human motor axons after hyperpolarization and to play a role in certain diseases associated with hyperexcitability, e.g., neuropathic pain and restless legs syndrome [35–37]. Our results indicate that h-currents can act as an ultra-fast and short-term adaptation mechanism in the motor control system that fine-tunes spinal excitability.

Coupling the motoneuron model with a skeletal muscle model could provide further insights into the functional role of postinhibitory excitation and its influence on, e.g., force steadiness and force variability [38]. Future modeling studies should further address the different possible causes of postinhibitory excitation each in isolation but in comparison to the other. Nevertheless, the most promising way to quantify the contribution of h-currents to postinhibitory excitation is *in-vitro* studies, also due to the uncertainty of the channel parameters in the model. This study pointed out that it is crucial to choose evaluation and analysis methods that can account for activation history.

## Summary and conclusion

We investigated how hyperpolarization-activated inward currents (h-currents) can contribute to postinhibitory excitation in human motor units. Using a computational model, we showed that h-currents can shorten interspike intervals in response to strong inhibitory stimuli and, thus, facilitate postinhibitory excitation. This effect is history-dependent and most pronounced in conditions with low firing rates and low noise, i.e., few other inputs. Furthermore, this study showed that intrinsic motoneuron properties must be considered for interpreting reflex responses. The presented PSF cluster method reveals history-dependent effects. Excitation in the first poststimulus interval after reciprocal inhibition was found in a significant portion of the analyzed human motor units. According to the simulation results, it can be speculated that the h-current serves as a modulatory mechanism that increases the excitability of motoneurons.

## Methods

This section describes the utilized computational model and data analysis techniques. The experimental data analyzed in this manuscript were collected in a previous study [7], and only a brief description of the experimental protocol is provided.

### Ethics statement

The data were acquired with the approval of the local ethical committee of the University Medical Center, Georg-August–University of Göttingen (approval date: 1/10/12). Each subject provided informed written consent before the experiments.

### Experimental data

Motor unit reciprocal inhibition data from a previous study [7] were used to investigate the incidence rate and strength of postinhibitory excitation activity *in vivo*. High-density surface EMG (HDsEMG) was recorded (sampling rate: 10240 Hz) from tibialis anterior muscles during steady isometric contraction at 10% and 20% of the maximum voluntary contraction (MVC). Stimulating the tibial nerve through monopolar stimulation electrodes elicited reciprocal inhibition on the tibialis anterior muscle. The metal pin anode and cathode electrodes were placed on the skin of the popliteal fossa to stimulate the nerve. The activity of individual motor units was identified by decomposing HDsEMG data using a blind source separation technique [39, 40]. Further details about the experimental protocol and analysis can be found in [7].

### Computational modeling

A single motoneuron was simulated using an electric circuit model based on the motoneuron model proposed in [18] and [38]. In short, the *in-silico* motoneuron consists of two compartments, i.e., the soma and a lumped dendrite. The soma compartment includes voltage-gated conductances that describe sodium and slow and fast potassium channels. Further, both compartments include an additional leakage conductance. For the present investigation, we added a voltage-gated h-channel conductance in both the soma and the dendrite compartment (Fig 7d). Accordingly, the membrane potential in each compartment can be described by the

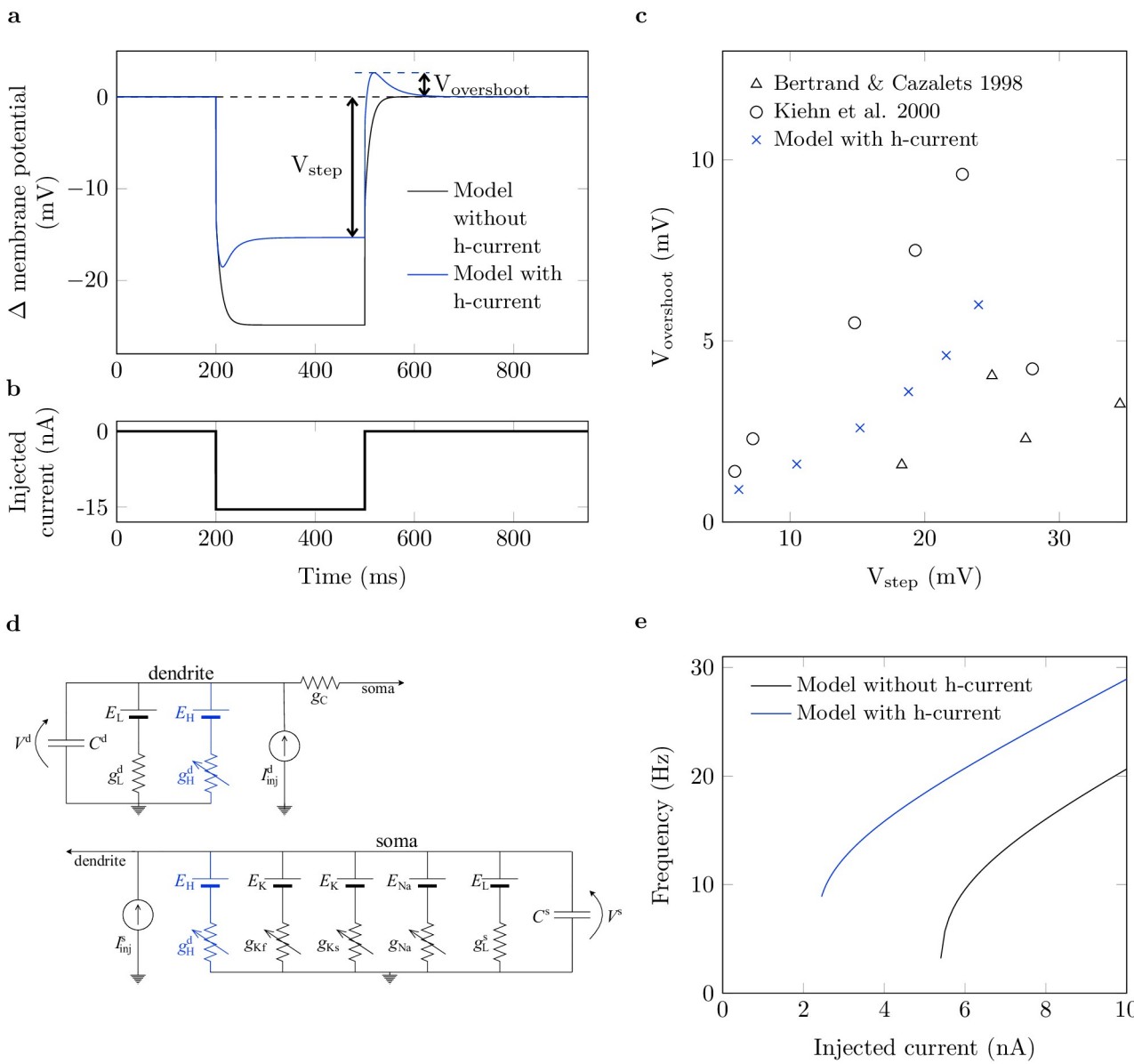

**Fig 7. Characteristic behavior of the computational motoneuron model.** (a): Membrane potential time course of the simulated motoneurons without (black) and with h-current (blue) in response to injection of the current step shown in (b). Membrane potential is given relative to the resting potential. (c): Steady-state membrane potential ($V_{step}$) vs. overshoot membrane potential ($V_{overshoot}$) of the simulated neuron (x) compared to data obtained from [26] (o) and [12] (△). (d): Equivalent electric circuit of the motoneuron model. Blue color highlights components added compared to the previous version of the model [18]. (e): Current-frequency relation for the model without (black) and with (blue) h-currents.

following differential equations:

$$C^d \frac{dV^d(t)}{dt} = -I_L^d - I_C^d - I_H^d + I_{inj}^d, \tag{1}$$

$$C^s \frac{dV^s(t)}{dt} = -I_L^s - I_C^s - I_{Na} - I_{Kf} - I_{Ks} - I_H^s + I_{inj}^s. \tag{2}$$

Therein, $V(t)$ is the membrane potential at time $t$ and $C$ is the membrane capacitance. The superscripts 's' and 'd' denote the soma and the dendrite compartment, respectively. The currents $I_{Na}$, $I_{Kf}$ and $I_{Ks}$ describe the flux of ions through sodium and fast and slow potassium channels, respectively. Further, $I_C$ describes the coupling current between the two compartments whereby $I_C^d = -I_C^s$. $I_L$ is a leakage current and $I_{inj}$ denotes currents injected into the compartments, e.g., by an external electrode. Currents are modeled and parameterized as described in [18] and [38]. H-channels were found to be expressed widely across motoneurons. Consequently, they are placed in both soma and dendrite compartments [16]. The mathematical description of the h-current and its parameters are provided in Eqs (3) to (5).

The h-current, $I_H$, is described by [19]:

$$I_H = \bar{g}_H \, p \, (V - E_H),\tag{3}$$

$$\frac{dp}{dt} = \frac{p_\infty - p}{\tau_H},\tag{4}$$

$$p_\infty = \frac{1}{1 + \exp[(V - V_{half})/V_{slope}]}.\tag{5}$$

Thereby, $E_H$ denotes the reversal potential of the h-conductance and $\bar{g}_H$ the maximum conductance. The transient behavior of the gating variable $p$ is determined by the voltage-independent time constant $\tau_H$, the half-maximum activation potential $V_{half}$ and a slope factor $V_{slope}$. Note that Eq (3) to Eq (5) refer to both compartments, i.e., $V$ corresponds to $V^s$ and $p$ to $p^s$ for the soma compartment and to $V^d$ and $p^d$ for the dendrite compartment.

## Model parameterization

Following a study by Duchateau and Enoka [41], we assumed that for tibialis anterior muscle and contraction strengths of less than 20% MVC, all recruited and recorded motor units are of slow type. Model parameters for different motor unit types were published by Cisi and Kohn [17] (see their Table 2). We chose the input resistance and the size parameters (radius and length) of both the soma and the dendrite compartment according to the mean values for slow-type motoneurons.

Since the h-channel was not considered by either Cisi and Kohn [17] or Negro and Farina [18], we used the implementation by Powers et al. [19] as a guide. Powers et al. [19] used experimental recordings by Larkman and Kelly [21] for the parameterization of the channel, and we adopted the time constant as well as the slope factor, i.e., $\tau_H = 50$ ms and $V_{slope} = 8$ mV. In contrast to Powers et al. [19], we assumed that persistent inward currents play a minor role in our study [25]. Thus, we adapted the reversal potential $E_H = 20$ mV and half-maximum activation potential $V_{half} = -20$ mV compared to the original implementation. The selected values agree with experimental data [21, 34]. Note that all potentials are given relative to the resting potential.

The above parameters determine the temporal dynamics of the h-current. The maximum conductance $\bar{g}_H$, which can be related with the density of h-channels in the membrane, determines the maximum amount of h-current that can flow across the membrane. Experimental current-clamp studies from Bertrand and Cazalets [12] and Kiehn et al. [26] were used to parameterize $\bar{g}_H$. Therefore, the simulated membrane potential trajectory in response to the injection of hyperpolarizing current steps was compared to *in-vitro* results. The simulated neuron shows the characteristic undershoot and overshoot at the beginning and end of the applied current step (Fig 7a and 7b), which is typically attributed to h-currents [11, 12, 21, 34].

For a quantitative comparison, the steady-state membrane potential just before the release of the current step, $V_{\text{step}}$, was compared to the maximum size of the membrane potential overshoot after the release of the current step, $V_{\text{overshoot}}$ (Fig 7a and 7b). The current step was applied for 300 ms to reach a steady membrane potential. With a maximum conductance of $\bar{g}_{\text{H}} = 2 \text{ ms cm}^{-2}$ the simulated neuron showed overshoot potentials of 0.84 mV to 6 mV in response to current injections that hyperpolarize the membrane potential by 6 mV to 24 mV. This is well within the range of values reported in Bertrand and Cazalets [12] and Kiehn et al. [26], i.e., overshoot potentials of 0.96 mV to 9.6 mV for potential steps of 5.9 mV to 34.5 mV (Fig 7c). Note that adding the h-current does not lead to subthreshold oscillations and thus preserves the integrator nature of the motoneuron model (Fig 7a).

Adding the h-current increases the neuron's resting potential due to an additional inflow of current, which was also reported by Powers et al. [19]. Consequently, the h-current model has a decreased rheobase, but the gain of the current-frequency relation is hardly affected (Fig 7e). We ensure comparability between the models by considering the simulated neurons at the same working point, defined by the frequency.

## Simulation of reciprocal inhibition

The applied simulation protocol aims to mimic the experimental procedure described in Section Experimental data. Repeated injections of inhibitory current pulses causing an IPSP imitate the stimulus delivered in the reflex experiment. The stimuli were applied 200 times with a random interstimulus interval of 1000 ± 100 ms. All simulations were performed with MATLAB R2021a (9.10.0.2015706). The motoneuron model is represented by a system of eight ordinary differential equations, which was solved with MATLAB's ode23 (single-step, explicit Runge-Kutta solver [42]) and an absolute and relative error tolerance of $1 \times 10^{-5}$.

The activity of the motoneuron model is driven by injected currents. To replicate the experimental protocol, the input is composed of up to three components: (i) inhibitory current pulses simulating reciprocal inhibition, (ii) a constant current representing the mean cortical drive to the neuron and which determines the contraction strength, (iii) additive noise representing all afferent and efferent inputs to the motoneuron that are not explicitly modeled.

Reciprocal inhibition (i) was simulated by injecting a current kernel representing the compound inhibitory postsynaptic current. The postsynaptic current $I_{\text{PSC}}$ is described by

$$I_{\text{PSC}} = -I_0 \, \frac{t}{\tau_{\text{PSC}}} \exp\left(1 - \frac{t}{\tau_{\text{PSC}}}\right), \tag{6}$$

with $t$ representing the time since the beginning of the current injection. The time constant $\tau_{\text{PSC}} = 4$ ms was fixed [43], while the amplitude $I_0$ ranged from 2.5 to 15 nA to cover a large range of IPSP strengths. The inhibitory postsynaptic current was applied for 20 ms, as this, for the chosen amplitudes, achieved the minimum firing rate at a time comparable to the experiment. The current-induced IPSPs usually last longer than 20 ms (Fig 3a and 3b).

The constant input into the neuron (ii) was chosen to obtain different baseline firing rates of approximately 10, 14, and 18 Hz. The noise component (iii) was composed of two parts, a zero-mean, band-pass filtered (15 Hz to 35 Hz) and a zero-mean low-pass filtered (<100 Hz) white noise [44, 45]. Thereby, the standard deviation of the noise input was scaled relative to the mean drive such that the band-pass filtered component accounts for 80% of the total standard deviation [46]. To investigate the effect of noisy inputs in the investigated reflex scenario, we applied different noise levels by scaling the total standard deviation of the summed noise input to 0%, 12.5%, and 25% of the constant input current (ii).

The input components (i) to (iii) were linearly summed and applied to the soma compartment, assuming they represent effective synaptic currents [18, 27]. To account for the delay caused by the nerve conduction velocity, a constant delay of 50 ms between IPSC application and membrane potential output was introduced.

## Data analysis

Motoneuron activity is described through discrete discharge times, i.e., a binary spike train. For the *in-vivo* experiment, spike trains were reconstructed by the decomposition of the recorded HDsEMG (for details, see [7, 39, 40]). In the simulation, the spike trains were directly obtained from the trajectory of the soma membrane potential.

The firing behavior of motoneurons before and after stimulation was analyzed using the peristimulus frequencygram (PSF) method (Fig 2a). The PSF shows the instantaneous discharge rates of motoneurons relative to the stimulus time [20]. Significant perturbations in discharge rate compared to baseline activity after the stimulus are likely related to the stimulating event. With increasing delay between the stimulus and the event, the probability that changes in discharge rate may reflect other processes increases [47–49]. The characteristics of the PSF response are assumed to depend both on the sign and the magnitude of a (reflex-induced) postsynaptic potential [49]. Although other factors may play a role, PSF is recognized as a valid method to quantitatively estimate *in vivo* the strength of inhibitory or excitatory postsynaptic potentials in motoneurons [47–49].

The strength of reciprocal inhibition and the consecutive excitation was determined by computing the cumulative summation of the PSF (PSF-cusum, Fig 2b) [50]. The PSF-cusum allows the measurement of subtle but consistent changes in the instantaneous discharge frequency. It was obtained by cumulatively summing the difference between the average baseline frequency ($-300$ ms $\leq$ time $< 0$ ms) and the instantaneous frequency value of each discharge.

The largest absolute deflection of PSF-cusum from zero during the baseline was determined as the significance threshold (error box) for reflex responses (Fig 2b, dashed lines). Troughs and peaks exceeding the error box represent significant inhibition and excitation responses. The response amplitude (inhibition or excitation) was defined as the difference between two turning points in the PSF-cusum (Fig 2b, arrows). The turning points correspond to the start and end points of the monotonous slope of a significant reflex response and were determined manually. We are aware that the duration of the inhibition is relevant. We did not consider the duration since we always used the same IPSC duration. The PSF-cusum was normalized by the number of delivered stimuli to compare amplitudes across subjects and trials with different numbers of stimuli. Consequently, PSF-cusum is shown in units of Hz/No. of Stim. It is worth noting that, in simulations where the membrane noise is not added, each deflection from zero directly relates to a change in the motoneuron's activity.

Recorded motor units were included in the analysis when at least 90 stimuli could be delivered. For simulated neurons, 200 stimuli were applied.

## Supporting information

**S1 Fig. Motoneuron model gating variables.** Motoneuron model gating variables for one interspike interval (duration 90.3 ms) of the model with (blue) and without (black) h-current. Shown are membrane potential (a), sodium channel (Na) activation gate (b), Na inactivation gate (c), fast potassium channels (Kf) activation gate (d), slow potassium channel (Ks) activation gate (e) and h-channel (H) activation gate (f). The gating variables are defined as described in [18] and [38].
(TIF)

**S2 Fig. Stimulus time-dependent duration of interspike intervals for different parameterizations of the h-channel.** Change of interspike interval duration ($\Delta$ ISI) over time of stimulus (IPSP) application with respect to the last discharge ($t_{stim}$). Default parameters are shown in black. In the simulations all parameters were fixed except for time constant $\tau_H$ (a), maximum conductance of h-current $\bar{g}_H$ (b), half-maximum activation potential $V_{half}$ (c) or reversal potential $E_H$ (d). Baseline frequency 10 Hz, no noise, inhibitory stimulus amplitude -10 nA. (TIF)

**S3 Fig. Stimulus time-dependent duration of interspike intervals for different parameterizations of the motoneuron model.** Change of interspike interval duration ($\Delta$ ISI) over time of stimulus (IPSP) application with respect to the last discharge ($t_{stim}$). Default parameters are shown in black. (a): Variation of the motoneuron size. FR-type motoneurons correspond to motoneurons with size-dependent parameters according to the mean values for FR-type neurons in [17]. Version A and B employ a maximum h-channel conductance of $\bar{g}_{H,A} = 2$ ms cm$^{-2}$ and $\bar{g}_{H,B} = 4$ ms cm$^{-2}$, respectively. (b): Influence of persistent inward currents (PICs) injected into the dendrite compartment of the motoneuron model. PIC dynamics according to [24]. Maximum PIC conductance 0.1 mScm$^{-2}$. Slow PIC: time constant 100 ms, reversal potential 140 mV. Fast PIC: time constant 1 ms, reversal potential 120 mV. Baseline frequency 10 Hz, no noise, inhibitory postsynaptic current amplitude -10 nA. (TIF)

## Author Contributions

**Conceptualization:** Laura Schmid, Thomas Klotz, Francesco Negro, Utku Ş. Yavuz.

**Investigation:** Laura Schmid.

**Methodology:** Laura Schmid.

**Writing – original draft:** Laura Schmid, Thomas Klotz, Utku Ş. Yavuz.

**Writing – review & editing:** Laura Schmid, Thomas Klotz, Oliver Röhrle, Randall K. Powers, Francesco Negro, Utku Ş. Yavuz.

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
