## [Decision Letter · Decision Letter 0]

13 Oct 2023

Dear PhD Yavuz,

Thank you very much for submitting your manuscript "Postinhibitory excitation in motoneurons can be facilitated by hyperpolarization-activated inward currents: a simulation study" for consideration at PLOS Computational Biology.

As with all papers reviewed by the journal, your manuscript was reviewed by members of the editorial board and by several independent reviewers. In light of the reviews (below this email), we would like to invite the resubmission of a significantly-revised version that takes into account the reviewers' comments.  All three reviewers appreciated the significance of the research, but all three indicated areas where additional simulations are required to improve the rigor of the research.  Please make sure that you add additional simulations to show robustness of parameters and add controls to show that other mechanisms cannot account for the results rior to re-submission.

We cannot make any decision about publication until we have seen the revised manuscript and your response to the reviewers' comments. Your revised manuscript is also likely to be sent to reviewers for further evaluation.

Sincerely,

Kim T. Blackwell, V.M.D., Ph.D.

Academic Editor

PLOS Computational Biology

Marieke van Vugt

Section Editor

PLOS Computational Biology

Reviewer's Responses to Questions

**Comments to the Authors:**

Reviewer #1: Additional comments are imbedded in the the uploaded PDF of the paper.

Summary:

This was a very interesting and enjoyable paper to read that proposed a novel mechanism of post-inhibition excitation from Ih to explain the excitatory rebound often seen following EMG/motor unit firing rate suppression from an IPSP. The simulations were well done and the data well presented. I only have a few comments for consideration and several edits to the manuscript to make the text more precise and more relatable to a physiologist.

Major comments.

1. There was no negative control performed in the simulation. That is, what would the PSF look like if an IPSP-EPSP from synaptic inputs was used in the model? Are there recordings of a hyperpolarized motoneuron that shows such a PSP profile? Perhaps the activation and inactivation characteristics of the Ih give a PSF profile that is closer to the human data than would a IPSP-EPSP profile from synaptic inputs? This could be elaborated on in the Discussion.

2. In the human data of Figure 1c, the amplitude of the post-inhibition excitation (peak of the PSF after 100 ms), is greater when the background firing rate is higher. This contrasts with the simulation results of Figure 4 where a lower drive produces a larger post-inhibition excitation. Can this discrepancy be explained?

3. Details of the simulated IPSPs need to be added to the text and figures. It appears in Figure 2 in the human data, the IPSP lasts for 50-60 ms (duration of the decreasing PSF CUSUM below significance level). Likewise, in Figure 3 the duration of the silent period plus the duration of the decrease in the PSF CUSUM appears to be produced by an IPSP of 50 to 60 ms. It would be beneficial to plot the simulated IPSP over the PSF in Figure 3 to relate the firing behaviour of the motoneuron to simulated IPSP and give IPSP parameters used for Figure 4. In Figure 5, the duration of the IPSP appears to be closer to 20 ms. Why was a 20 ms IPSP chosen and would similar results be obtained if an IPSP of 50-60 ms was used?

Reviewer #2: This is an interesting and well written study about motor neuron physiology and the role of intrinsic neuron's property on postinhibitory excitation following reciprocal inhibition. The paper combines human experiments and computer simulations to show some evidence on the involvement of H current on postinhibitory excitation. Despite the relevance of the topic, the opinion of this reviewer is that the current investigation neglects important biophysical aspects and thereby precludes a definite conclusion about the actual mechanisms behind postinhibitory excitation observed in human motor units.

- The authors discuss other potential mechanisms that would be involved in postinhibitory excitation, and they stated that they cannot rule out the involvement of some of them in behavior from human motor units. However, after reading the paper and the rationale provided by the authors one can have the idea that postinhibitory excitation in humans should be attributed to Ih. My opinion is that the authors don't have a strong evidence in favor of this argument and, therefore, they should revise the text to make clear that simulation results cannot ensure the real mechanism in human motor units involves Ih.

- In line with my previous comment, Bertrand and Cazalets showed evidence that postinhibitory excitation is mediated by Ca++-activated K+ current. The authors did not explore the role of Ca++-activated K+ current and other potential mechanisms (such as the deinactivation of K+ conductances). The Cisi-Kohn's model does not represent these molecular mechanisms, and therefore the simulation results are not comprehensive enough to be linked to a specific behavior observed experimentally.

- Manuel et al. (2007) showed that Ih causes the resonant behavior of the motoneuron membrane. However, they found that Ih (and resonant behavior) is typical of motoneurons innervating fast-contracting muscle fibers and that slow motoneurons do not exhibit resonant behavior. Also, they found that the resonant behavior is primarily affected by Ca++ and Na+ persistent inward currents (PICs). My point is that the authors neglected the influence of PICs on this postinhibitory excitation. I am not confident that the lack of effect of PICs on reciprocal inhibition (as stated in the paper) is sufficient to adopt a model without PICs and their potential influence on the behavior the authors are trying to represent. Additionally, the authors described an S-type motoneuron, which was experimentally less susceptible to resonate and exhibit the voltage “sag” (overshoot).

- One weakness of the study is the lack of a proper sensitivity analysis regarding model's parameters. The authors varied some input parameters, such as excitation level, noise level, and inhibition amplitude. But it is not clear how changes in morphological properties or other electrophysiological properties would affect the postinhibitory excitation.

- After including the Ih current the f-I curve of the motor neuron changed considerably. The authors argued that the gain was maintained, but it is evident that the model with Ih is much more excitable than the model without Ih. The curve is upward shifted, so that to maintain the same mean discharge rate the neuron with Ih will require a much lower injected current. How the magnitude of the input excitation would affect Ih and the other ionic currents? Additionally, with the inclusion of Ih the electrophysiological properties of the motor neuron seem very different from a typical anesthetized S-type motoneuron (see Zengel's data). What would happen if you adjust the other ionic currents so that the model with Ih would resemble the f-I curve of a typical S-type motor neuron? Also, other electrophysiological properties are important to validate the model, such as the AHP amplitude and duration, rheobase, etc.

- The condition where postinhibitory excitation was frequently observed in simulations seems quite stringent (and probably non-physiological). Absence or low noise is unlikely in the motor system. What happens if the neuron is discharging closer to its threshold? Do you think the role of noise is different in the latter condition? Again, this comment is in line with my previous argument that the simulations are not comprehensive enough to guarantee that the mechanism behind postinhibitory excitation was fully explored.

- The magnitude of postinhibitory excitation obtained from computer simulations is very different from those obtained in the experiments. The authors should discuss the potential causes for these quantitative differences. Do you think other mechanisms would be involved? Can you guarantee that the set of parameters are optimally chosen?

Reviewer #3: This study aims to better understand the processes behind a rebound of motoneuron firing rate after receiving inhibitory synaptic inputs. This rebound has been observed in vivo in humans on motoneuron firing rates extracted during isometric contractions by combining high-density electromyography recordings and decomposition algorithms based on blind-source separation.

As it is not possible to directly record in vivo in humans the synaptic inputs and inward currents causing these changes in firing rate, the authors used a biophysical model of motor neurons to directly test their hypotheses and reproduce the motoneuron firing activity observed in experiments. Their hypothesis is that ‘h-currents’ transiently change the excitability of the motor neuron, which in turn increase the probability of firings. The study appears well-conducted, with the use of classic biophysical models developed and refined in parallel by several teams (Heckman Powers Binder…, Cisi Kohn Elias…, Negro Farina Dideriksen…) and the conclusions are convincing. I have therefore only minor questions and comments.

Minor comments:

1.

Motoneuron firing rates have been extracted while the participants were performing a force-matched task with visual feedback. I was wondering whether the authors observed a short drop in force with the nerve stimulation followed by a sharp increase of force to correct the error in tracking and go back to the target? If yes, could the observed rebound in firing rate be caused by a transient change in net synaptic drive proportional to the force, with a pattern of firing rate matching the force profile?

In general, adding variations in force on the plots and the results would be appreciated to convince the reader that your rebound is probably due to other factors, such as inward currents.

2.

The duration of the contractions during the experiment is quite long (250s), at non-negligeable level of force (10 to 20%MVC). Beside fatigue, it is standard to observe a slow decrease in firing rate during long plateaus of force due to accommodation or adaptation phenomenon (Revill & Fuglevand, 2011, J Neurophysiol). Such variations are link to slow changes in the intrinsic properties of motor neurons and the inactivation of ionic channels. I was therefore wondering whether you observed a time effect on the rebound after the stimuli, with different responses at the beginning and the end of the contraction? Such changes could be linked to other factors than those tested by the authors.

3.

All the simulations are performed on a single motor neuron with fixed properties such as the rheobase, soma/dendrites radius, length, and resistance. I understand this choice for the sake of simplicity, but it would be interesting to have a sense on the impact of the size of the motor neuron on the presence of a rebound or not after an inhibitory stimulus. Moreover, having an idea on the effect of this phenomenon at the population level would help to infer potential functional consequences.

4.

It is sometime difficult to understand the analyses and the results without going to the method at the end of the manuscript. This is annoying when the results come first in the paper. I would recommend adding more details about the methods in the result section to improve the flow of the story, and simplify the life of the reader.

Specific comments:

Results:

L. 109 the ‘e’ is missing in Therefore

**Have the authors made all data and (if applicable) computational code underlying the findings in their manuscript fully available?**

Reviewer #1: Yes

Reviewer #2: **No: **They stated that all data will be available upon paper acceptance.

Reviewer #3: **No: **

PLOS authors have the option to publish the peer review history of their article (what does this mean?). If published, this will include your full peer review and any attached files.

Reviewer #1: **Yes: **Monica Gorassini

Reviewer #2: No

Reviewer #3: **Yes: **Simon Avrillon
---

## [Decision Letter · Decision Letter 1]

19 Dec 2023

Dear PhD Yavuz,

Thank you very much for submitting your manuscript "Postinhibitory excitation in motoneurons can be facilitated by hyperpolarization-activated inward currents: a simulation study" for consideration at PLOS Computational Biology. As with all papers reviewed by the journal, your manuscript was reviewed by members of the editorial board and by several independent reviewers. The reviewers appreciated the attention to an important topic. Based on the reviews, we are likely to accept this manuscript for publication, providing that you modify the manuscript according to the review recommendations.  One of the reviewers still had a question about membrane potential trajectory after the IPSP, that needs to be clarified.

Sincerely,

Kim T. Blackwell, V.M.D., Ph.D.

Academic Editor

PLOS Computational Biology

Marieke van Vugt

Section Editor

PLOS Computational Biology

Reviewer's Responses to Questions

**Comments to the Authors:**

Reviewer #1: Review is uploaded as an attachment.

Reviewer #2: I would like to thank the authors for answering all my questions. I consider that all my concerns have been sufficiently addressed in the revised version of the manuscript, including the figures from new simulations (supplementary material). I don't have any additional suggestion.

Reviewer #3: I thank the authors for their thorough review and their responses to the comments of other reviewers. The manuscript has become much clearer, and the inclusion of methodological information in the results section improves the flow of the document. Additionally, the provision of data and codes on the server is a valuable contribution to our community.

**Have the authors made all data and (if applicable) computational code underlying the findings in their manuscript fully available?**

Reviewer #1: Yes

Reviewer #2: Yes

Reviewer #3: Yes

PLOS authors have the option to publish the peer review history of their article (what does this mean?). If published, this will include your full peer review and any attached files.

Reviewer #1: **Yes: **Monica Gorassini

Reviewer #2: **Yes: **Leonardo Abdala Elias

Reviewer #3: **Yes: **Simon Avrillon

Figure Files:

Data Requirements:

Reproducibility:

References:

---

## [Decision Letter · Decision Letter 2]

9 Jan 2024

Dear PhD Yavuz,

We are pleased to inform you that your manuscript 'Postinhibitory excitation in motoneurons can be facilitated by hyperpolarization-activated inward currents: a simulation study' has been provisionally accepted for publication in PLOS Computational Biology.

Best regards,

Kim T. Blackwell, V.M.D., Ph.D.

Academic Editor

PLOS Computational Biology

Marieke van Vugt

Section Editor

PLOS Computational Biology

Reviewer's Responses to Questions

**Comments to the Authors:**

Reviewer #1: Thank you for including the IPSP information as this ultimately determines the profile of the PSF. It would make the manuscript clearer if you provide the mean duration of the inhibition response (IPSP) for the human motor unit data on page 9, para 2 of the red line version, along with the provided amplitude information - ie., duration of decrease in the PSF cusum which in Figure 2 looks to be around 60ms. Please also indicate for the simulated data on Page 9, line 130, that you are applying a 20 ms IPSC that results in a ~50 ms IPSP. It was the lack of IPSC/IPSP information at this point in the paper that got me confused.

**Have the authors made all data and (if applicable) computational code underlying the findings in their manuscript fully available?**

Reviewer #1: Yes

PLOS authors have the option to publish the peer review history of their article (what does this mean?). If published, this will include your full peer review and any attached files.

Reviewer #1: **Yes: **Monica Gorassini

---

## [Editor Report · Acceptance letter]

17 Jan 2024

PCOMPBIOL-D-23-01428R2 

Postinhibitory excitation in motoneurons can be facilitated by hyperpolarization-activated inward currents: a simulation study

Dear Dr Yavuz,

I am pleased to inform you that your manuscript has been formally accepted for publication in PLOS Computational Biology. Your manuscript is now with our production department and you will be notified of the publication date in due course.

With kind regards,

Bernadett Koltai
